# The Present and Future of Neoadjuvant and Adjuvant Therapy for Locally Advanced Gastric Cancer

**DOI:** 10.3390/cancers15164114

**Published:** 2023-08-15

**Authors:** Anna S. Koerner, Ryan H. Moy, Sandra W. Ryeom, Sam S. Yoon

**Affiliations:** 1Division of Surgical Oncology, Department of Surgery, Columbia University Irving Medical Center, New York, NY 10032, USA; 2Vagelos College of Physicians and Surgeons, Columbia University, New York, NY 10032, USA; 3Division of Hematology/Oncology, Department of Medicine, Columbia University Irving Medical Center, New York, NY 10032, USA; 4Division of Surgical Sciences, Department of Surgery, Columbia University Irving Medical Center, New York, NY 10032, USA

**Keywords:** gastric cancer, neoadjuvant therapy, adjuvant therapy, locally advanced, immunotherapy, targeted therapy, biomarkers

## Abstract

**Simple Summary:**

Gastric cancer is a deadly disease with worldwide prevalence that is often diagnosed at late stages. About two-thirds of patients in Western countries will present with locally advanced gastric cancer (LAGC). When patients are diagnosed with LAGC, they frequently undergo surgery and perioperative chemotherapy. However, the most effective multimodality treatment regimen for LAGC has yet to be determined. In aiming to improve outcomes, current trials are examining immunotherapies and targeted therapies based on a growing understanding of the unique molecular characteristics and subtypes of LAGC. This review summarizes current and future medical therapies for LAGC.

**Abstract:**

Gastric cancer is a highly prevalent and lethal disease worldwide. Given the insidious nature of the presenting symptoms, patients are frequently diagnosed with advanced, unresectable disease. However, many patients will present with locally advanced gastric cancer (LAGC), which is often defined as the primary tumor extending beyond the muscularis propria (cT3-T4) or having nodal metastases (cN+) disease and without distant metastases (cM0). LAGC is typically treated with surgical resection and perioperative chemotherapy. The treatment of LAGC remains a challenge, given the heterogeneity of this disease, and the optimal multimodal treatment regimen may be different for different LAGC subtypes. However, many promising treatments are on the horizon based on knowledge of molecular subtypes and key biomarkers of LAGC, such as microsatellite instability, HER2, Claudin 18.2, FGFR2, and PD-L1. This review will expand upon the discussion of current standard neoadjuvant and adjuvant therapies for LAGC and explore the ongoing and future clinical trials for novel therapies, with information obtained from searches in PubMed and ClinicalTrials.gov.

## 1. Introduction

Gastric cancer (GC) represents the world’s fifth most common cancer and fourth leading cause of cancer death, with 1.1 million new cases and 770,000 deaths estimated in 2020 [1,2]. The incidence of GC is higher in males than in females in both developed and developing countries [1]. Incidence rates are highest in Eastern and Central Asia and Latin America; the average incidence rate of GC in Eastern Asia is 32.1 per 100,000 in males and 13.2 per 100,000 in females [3]. The incidence of GC peaks in the seventh and eighth decades of life [4]. Risk factors for non-cardia GC are *Helicobacter pylori* (*H. pylori*) infection, high-salt diets, smoking, prior gastric surgery, and family history [4]. Cardia GC is associated with gastroesophageal reflux disease (GERD) and obesity [5]. The incidence of non-cardia GC has decreased worldwide over the past 50 years; however, cardia GC has increased sevenfold, especially in developed countries, and an unexpected increase in the incidence of GC has been seen in younger age groups (<50 years old) [6,7,8]. The overall decrease in non-cardia GC may be due to improved *H. pylori* treatments and advancements in food hygiene in recent decades [9].

The most common presenting symptoms of GC are non-specific weight loss, dyspepsia, abdominal pain, anorexia, and dysphagia [10]. Given the insidious nature of these symptoms, a majority of GC patients present with advanced or metastatic disease [11]. Therefore, mortality rates are high in GC, with patients presenting with unresectable or metastatic GC having a median survival of about one year with medical treatment [12]. Approximately two-thirds of patients in Western countries will present with locally advanced gastric cancer (LAGC), which is usually defined as the primary tumor extending beyond the muscularis propria (cT3-T4) or having nodal metastases (cN+) disease and without distant metastases (cM0) [11,13]. In contrast, the majority of patients in Japan and Republic of Korea present with early disease due to nationwide screening programs [14,15].

## 2. Current Perioperative Chemotherapy for LAGC

Although surgery is the major component of curative-intent treatment of LAGC, there is a significant risk of recurrence with surgery alone: around 65–75% in Western countries and under 50% in Japan and Republic of Korea within five years [16,17,18,19,20]. Therefore, the treatment of LAGC often includes the addition of medical therapy. The rationale for perioperative chemotherapy is to downstage the tumor to facilitate resection, to eradicate micrometastatic disease, and to deliver therapy prior to gastrectomy when it is often better tolerated [21,22]. Since the publication of the landmark MAGIC trial in 2006, perioperative chemotherapy is now often recommended for LAGC [16]. The MAGIC trial randomized patients with resectable gastric, gastroesophageal junction (GEJ), or lower esophageal adenocarcinomas to either three cycles of epirubicin, cisplatin, and fluorouracil (ECF) before and after surgery or surgery alone. The perioperative chemotherapy group had a significantly improved overall survival (OS) (5-year OS 36% versus 23%) compared to that of the group undergoing surgery alone. Another randomized controlled trial (RCT), the FNCLCC/FFCD multicenter phase III trial, found that, in patients with resectable GEJ/GC, the use of perioperative chemotherapy with fluorouracil and cisplatin greatly improved the disease-free survival, OS, and curative resection rate when compared to surgery alone (5-year OS 38% versus 24%) [23]. The FLOT-AIO trial, published in 2019, revised the standard perioperative chemotherapy regimen used for LAGC. This trial randomized patients with resectable gastric or GEJ adenocarcinoma (cT2 or higher and/or cN+ resectable tumors with no evidence of distant metastases) to the MAGIC regimen chemotherapy and surgery or four cycles of fluorouracil, oxaliplatin, and docetaxel (FLOT) before and after surgery [24]. The median OS (50 months versus 35 months) and the three-year OS (57% versus 48%) were significantly longer in the FLOT group, and pathological complete regression was 6% in the MAGIC group and 16% in the FLOT group.

Unlike for perioperative chemotherapy, the role of neoadjuvant and adjuvant chemoradiation remains unclear in LAGC. The number of RCTs investigating the efficacy of preoperative chemoradiation in resectable LAGC remains limited. However, a meta-analysis of seven RCTs of neoadjuvant chemoradiation therapy (NCRT) showed higher rates of pathologic complete response (pCR), R0 resection, and longer 1-year and 3-year survival rates compared to patients who underwent neoadjuvant chemotherapy [25]. Notably, four of the seven RCTs did not report survival rates. Several non-randomized studies have shown that, in patients with LAGC, NCRT is more likely to achieve pCR compared to neoadjuvant chemotherapy but with controversial survival benefits [26,27,28,29]. The ARTIST II randomized trial compared three adjuvant regimens, oral S-1, S-1 plus oxaliplatin (SOX), and SOX plus S-1 chemoradiation (SOXRT), in patients with stage II/III node-positive GC after gastrectomy and D2 lymphadenectomy. They found that SOX and SOXRT were similarly effective in prolonging disease-free survival compared to S-1 monotherapy [30]. There was no significant reduction in recurrence with the addition of radiation to SOX.

Many promising new therapeutics are on the horizon for the treatment of LAGC. This review will discuss the growing understanding of the molecular characteristics of GC as well as current and ongoing major studies that will inform future directions of perioperative treatment.

## 3. Molecular Subtypes and Biomarkers of GC

GC is a heterogenous disease. Over the past decade, an in-depth understanding of the molecular characteristics of GC has led to the advent of novel diagnostic and prognostic biomarkers as well as the development of targeted therapies. With precision medicine and an increasing ability to predict responders to various therapies, neoadjuvant and adjuvant therapies can be better utilized in GC patients.

In 2014, The Cancer Genome Atlas (TCGA) outlined a molecular classification system separating GC into four subtypes: tumors with high microsatellite instability (MSI-H), tumors that are Epstein–Barr virus positive (EBV+), genomically stable (GS) tumors, and tumors with chromosomal instability (CIN) [31]. These subtypes each have distinct genomic features, which may serve as targets for therapeutic agents. In 2015, the Asian Cancer Research Group published their own molecular classification of GC with associated prognostic differences: tumors with microsatellite instability (MSI), mesenchymal tumors (microsatellite-stable tumors with epithelial-to-mesenchymal transition phenotype (MSS/EMT)), and non-EMT MSS tumors that are p53+ and p53− [32]. Patients with MSS/EMT-type GC were younger and had the worst prognosis compared to patients with the other subtypes. In contrast, patients with MSI-H tumors had the best prognosis; the MSS/p53+ genotype was closely linked to EBV+ status, and patients with MSS/p53− tumors had a better prognosis than those with MSS/p53+ tumors.

MSI-H tumors have a high tumor mutational burden and are deficient in one or more mismatch repair proteins, i.e., mismatch repair deficient (dMMR). dMMR is a surrogate for the MSI-H phenotype. Polymerase chain reaction can be used to detect amplified microsatellite loci and is more accurate than IHC, though more expensive [33]. MSI-H tumors are associated with older age and tend to occur in the distal stomach [34]. About 10–20% of GCs are MSI-H [35,36]. MSI-H tumors tend to have fewer lymph node metastases and better OS [37,38]. In addition, MSI-H GCs frequently express immune checkpoint molecules like PD-L1, making them possibly eligible for treatment with immune checkpoint inhibitors (ICIs) [39]. MSI-H GCs can be identified using immunohistochemistry (IHC) with antibodies directed at mismatch repair proteins: MLH1, PMS2, MSH2, and MSH6 [38].

The EBV+ subtype of GC accounts for 10% of cases and frequently harbors PIK3CA and ARID1A mutations as well as amplification of JAK2, PD-L1, and PD-L2 and DNA hypermethylation [31]. Testing for EBV infection is best implemented by EBV encoding region (EBER) in situ hybridization, which would show a nuclear signal [40]. EBV-associated GC is more common in younger men and occurs primarily in the proximal stomach or postsurgical gastric stump [41]. The EBV+ subtype of GC generally has a more favorable prognosis and fewer lymph node metastases than non-EBV GC [42]. In addition, given that the EBV+ subtype of GC often is associated with modulation of the tumor microenvironment (TME) to evade the host immune response (such as by upregulation of PD-L1 expression by *CD274* focal amplification and *IFN*-γ-mediated signaling, causing immunosuppression), immune checkpoint blockade arose as a promising avenue of treatment [43,44,45].

HER2 gene amplification occurs in 10–30% of GCs, which results in overexpression of HER2, a receptor tyrosine kinase [31]. HER2 status is mainly assessed through IHC and/or fluorescence in situ hybridization (FISH). Given its relatively low cost and ease of performance, IHC is typically used first to test for HER2 overexpression. A 3+ result is considered positive. If the results are equivocal (2+), FISH is performed [46]. HER2+ status is prognostic of poorer outcomes in GC, so identification is critical and may predict which patients will benefit from the addition of trastuzumab or other HER2-directed therapies to their chemotherapy regimen [26,47].

Finally, PD-L1 IHC is routinely performed to help predict potential benefit from immunotherapy. In addition, the EBV+ and MSI-H subtypes are associated with the overexpression of PD-LI in tumor, stromal, and immune cells [31]. The combined positive score (CPS) quantifies PD-LI expression in tumor and tumor-associated immune cells, where CPS ≥ 1 is considered positive [48]. A higher score correlates with an increased probability of clinical benefit from PD-L1 inhibition [49]. An increasing number of studies have been conducted to investigate the efficacy of immunotherapy for LAGC patients, which will be discussed later in this article.

## 4. Immune Checkpoint Blockade for MSI-H LAGC

As mentioned above, MSI-H LAGC is associated with an improved prognosis [16,31,50]. In the MAGIC trial, which established perioperative chemotherapy as the standard of care for non-Asian patients with LAGC, patients with MSI-H tumors had an improved prognosis compared to patients with microsatellite-stable (MSS) tumors in the surgery-only group (hazard ratio (HR) 0.35, 95% confidence interval (CI) 0.11–1.11, *p* = 0.08). Perioperative chemotherapy and surgery in the MSI-H subset of patients was associated with a worse OS compared to surgery alone (HR = 2.22, 95% CI 1.02–4.85, *p* = 0.04) [16,51]. In the CLASSIC trial, which found that adjuvant capecitabine and oxaliplatin after curative D2 gastrectomy is effective in Asian LAGC patients, patients with MSI-H tumors did not have a survival benefit when adjuvant chemotherapy was added to surgery (5-year disease-free survival 83.5% versus 85.7%, *p* = 0.931) [50,52]. A pooled meta-analysis of four neoadjuvant/adjuvant chemotherapy trials (MAGIC, CLASSIC, ARTIST, ITACA-S) found that only the non-MSI-H LAGC patients benefited from chemotherapy and surgery compared to surgery only, with a five-year OS of 62% versus 53% (HR = 0.75, 95% CI 0.60–0.94) [53]. Thus, neoadjuvant/adjuvant chemotherapy is likely ineffective for MSI-H LAGC.

However, there may be a role for neoadjuvant or adjuvant immunotherapy, especially in patients with MSI-H tumors. The TME plays a key role in GC progression through immune metabolic reprogramming and alteration of immune cells to evade host defenses [54]. The immune TME in GC contains T and B lymphocytes, natural killer (NK) cells, macrophages, and neutrophils [55]. These cells can promote or inhibit tumor immunity. GC exerts a variety of mechanisms to evade the host immune system. For instance, tumors can release anti-inflammatory cytokines such as IL-10 and TGF-β, which can cause the recruitment of regulatory T cells (Tregs) and inhibit the antitumor effect of lymphocytes [55,56]. Tregs can cause immune suppression through multiple mechanisms, including CTLA-4 expression to inhibit antigen-presenting cells [57]. In addition, GCs themselves can express PD-L1, which interacts with PD-1 on the cell surface of T cells, causing anergy and/or apoptosis of T cells [58]. With this knowledge, ICIs have become a promising area of therapy for many tumor types including GC.

Several studies show an improved response to ICIs when combined with chemotherapy in a variety of cancers, such as non-small-cell lung cancer, mesothelioma, and renal cell carcinoma [59,60,61]. This synergistic effect is thought to be due to chemotherapy activating an endogenous antitumor immune response, causing the enhancement of co-stimulatory molecules like CD80 and CD86 and the downregulation of PD-L1, and inducing immunogenic tumor cell death [62]. The current Food and Drug Administration (FDA)-approved ICIs target two key T-cell signaling pathways: PD-L1/PD-1 and CTLA-4. Although PD-L1 expression in GC has been associated with poorer outcomes, anti-PD-L1 antibodies (e.g., atezolizumab, avelumab, durvalumab) and anti-PD-1 antibodies (e.g., nivolumab, pembrolizumab) have been shown to improve OS in patients with advanced or metastatic GC [63,64]. CTLA-4 inhibitors such as tremelimumab and ipilimumab have also been explored as options for the treatment of GC. The KEYNOTE-062 phase III trial randomized patients to pembrolizumab alone, pembrolizumab and chemotherapy (5-FU and oxaliplatin), and chemotherapy alone in patients with advanced GC and PD-L1 CPS ≥ 1. That study found that pembrolizumab was as effective as chemotherapy with fewer treatment-related adverse effects [65]. Overall, pembrolizumab plus chemotherapy was not more effective than chemotherapy alone [65]. Post hoc analysis of the KEYNOTE-062 trial, along with the KEYNOTE-059 and KEYNOTE-061 trials, found a significant benefit for pembrolizumab in the subset of patients with MSI-H tumors [66]. One meta-analysis analyzed the RCTs that studied first-line advanced GC treatment with chemotherapy plus ICIs vs. chemotherapy alone [67]. That analysis found a significant reduction in the risk of death and progression in those treated with combination chemotherapy plus ICIs, across patients with PD-L1 CPS ≥ 10 and CPS ≥ 1, but a larger reduction in risk of death and progression was seen in patients with CPS ≥ 10.

Given the efficacy of PD-1/PD-L1 inhibition in MSI-H advanced GC, the question arose of whether similar effects would be seen in MSI-H LAGC. The GERCOR NEONIPIGA phase II trial evaluated the pathological complete response (pCR) rate after perioperative chemotherapy in GC/gastroesophageal junction adenocarcinoma (GEJC) with dMMR/MSI-H status [68]. Specifically, 32 patients with cT2-T4/NX/M0 resectable dMMR/MSI-H GC/GEJC underwent neoadjuvant nivolumab and ipilimumab for four cycles followed by surgery and adjuvant nivolumab for nine cycles [68]. Overall, 59% of tumors had complete regression with no residual tumor (TRG1a) and 14% had less than 10% residual tumor (TRG1b) [68,69]. At the most recent follow up of those patients who achieved Becker TRG 1a/1b, no patients had died or had recurrence. The authors concluded that neoadjuvant and adjuvant nivolumab and ipilimumab was safe and feasible in patients with MSI-H LAGC [68].

The phase II INFINITY trial enrolled patients with MSI-H cT2 or greater (any N, M0) GC/GEJC (n = 15) and treated these patients with tremelimumab (anti-CTLA-4) and durvalumab (anti-PD-L1) for one cycle followed by durvalumab every four weeks for two cycles followed by surgery [70]. Ultimately, 60% of patients achieved pCR, and 80% had <10% viable tumor. Notably, only one out of seven patients with T4 tumors achieved pCR [70]. In summary, neoadjuvant and adjuvant immunotherapy with anti-PD-L1 and anti-CTLA-4 agents has demonstrated promising tumor-eradicating activity in patients with dMMR/MSI-H LAGC.

## 5. Immune Checkpoint Blockade Plus Chemotherapy for Non-MSI-H LAGC

Although immune checkpoint blockade has been shown to be effective in MSI-H LAGC, most LAGCs are MSS [36]. ICIs in combination with chemotherapy may be effective in non-MSI-H LAGC based on studies in advanced patients. The CheckMate-649 phase III trial found that, patients with advanced GC/GEJC treated with nivolumab, an anti-PD-1 antibody, obtained a survival benefit when combined with chemotherapy compared to chemotherapy alone in patients with tumors with PD-L1 CPS ≥ 1 and ≥5 [49]. This led to the FDA approval of nivolumab in April 2021 for use in advanced GC/GEJC regardless of PD-L1 CPS, though the National Comprehensive Cancer Network (NCCN) guidelines specify their category 1 designation for CPS ≥ 5 [71]. The CheckMate-577 trial showed that nivolumab can also prolong survival compared to placebo in the adjuvant setting in patients with resected stage II and III esophageal or GEJC who also received neoadjuvant chemoradiotherapy [72]. Recent data from key phase III RCTs performed in China, Rationale-305 and ORIENT-16, show an OS benefit in patients with PD-L1+ unresectable GC/GEJC treated with PD-L1 inhibitors in combination with chemotherapy compared to placebo and chemotherapy [73,74]. Rationale-305 found that the PD-L1+ group (defined as CPS ≥ 5) treated with tislelizumab and chemotherapy had improved progression-free survival (PFS) and median OS (17.2 vs. 12.6 months, one-sided *p*-value = 0.0056) [74]. ORIENT-16 demonstrated superior OS with sintilimab and chemotherapy in all patients regardless of PD-L1 status (31.9% risk reduction, *p* < 0.0001), with a stronger reduction in patients with CPS ≥ 5 (41.3% risk reduction, *p* < 0.0001) [73]. ATTRACTION-4, a phase II-III multicenter trial across Japan, Republic of Korea, and Taiwan, randomized patients with unresectable advanced or recurrent GC/GEJC to either nivolumab with chemotherapy or placebo and chemotherapy; they found that the addition of nivolumab significantly improved PFS (10.45 vs. 8.34 months, *p* = 0.0007) but not OS [75].

Several studies are underway to investigate the efficacy of ICIs with neoadjuvant and/or adjuvant chemotherapy for non-MSI-H LAGC. A single-arm phase II trial (NCT0291816) recruited 34 patients with LAGC/GEJC; these patients were treated with capecitabine and oxaliplatin (CAPOX) before and after surgery as well as one cycle of pembrolizumab immediately before surgery followed by twelve months of maintenance pembrolizumab [76]. It was found that 20.6% of patients achieved pCR, surpassing the target pCR of 15% [76]. DANTE is a multicenter phase II trial that is comparing perioperative atezolizumab (anti-PD-L1) and FLOT chemotherapy against FLOT alone in patients with operable LAGC regardless of PD-L1 status (≥cT2 or N+ GC/GEJC) [77]. The preliminary results show an increase in pathological regression rates in patients who received both atezolizumab and FLOT, particularly in patients with higher PD-L1 expression (overall pT0 23% vs. 15%, pN0 68% vs. 54%). The estimated accrual completion date is 2025. In addition, the KEYNOTE-585 trial is underway to investigate the efficacy and safety of neoadjuvant and adjuvant pembrolizumab and chemotherapy (either FLOT, cisplatin and 5-FU, or cisplatin and capecitabine) compared to placebo and chemotherapy in patients with stage II-IVa GC or GEJC [78]. The primary endpoints are OS, event-free survival, and pCR rate [78]. KEYNOTE-585 is estimated to complete accrual in June 2024 and will hopefully further elucidate the role of adjuvant immunotherapy in non-MSI-H LAGC. Finally, the MATTERHORN trial is currently recruiting around 900 patients with resectable stage II or higher GC/GEJC. These patients will be randomly assigned to receive either neoadjuvant durvalumab or placebo in combination with FLOT chemotherapy, followed by either adjuvant durvalumab or placebo monotherapy after surgical resection [79]. Durvalumab in combination with FLOT has been shown to have the potential to improve outcomes in two smaller clinical studies in patients with advanced GC/GEJC, hence leading to the development of MATTERHORN [80,81]. Expected accrual completion is in 2025.

Recent studies suggest that the timing of immunotherapy with chemotherapy may be critical the in treatment of GC. ATTRACTION-5 is a phase III trial of Asian patients with stage III GC/GEJC randomized to receive either nivolumab with adjuvant chemotherapy or placebo with adjuvant chemotherapy [82]. There was no improvement in OS, and the primary endpoint of PFS was not met (HR = 0.90, *p* = 0.4363) [82]. A phase II randomized trial (NCT04250948) evaluated the efficacy of the addition of toripalimab to perioperative chemotherapy in patients with LAGC/GEJC [83]. In the group receiving PD-L1 inhibitors, toripalimab was added preoperatively and as a maintenance adjuvant therapy. Patients in the toripalimab plus chemotherapy arm achieved a higher proportion of pCR (24.1% vs. 9.3%, *p* = 0.039) and TRG 0/1 (44.4% vs. 20.4%, *p* = 0.009) than patients in the chemotherapy arm [83]. Although these studies were conducted in different populations, together they suggest that perioperative immunotherapy may be more effective than adjuvant only when combined with chemotherapy. The upcoming results of MATTERHORN and KEYNOTE-585 will further inform our understanding of the role of immunotherapy in LAGC.

## 6. HER2 Blockade for HER2+ LAGC

As mentioned earlier, 10–30% of GC/GEJCs exhibit HER2 amplification, which promotes tumor progression through downstream pathways such as PI3K/Akt/mTOR and MAPK [31,84]. HER2 overexpression was initially recognized in breast cancer and now has been recognized in other cancers such as colon, bladder, and GC [85]. HER2 expression in GC has been associated with more aggressive disease and poorer outcomes [86,87].

In the past 20 years, five HER-2-targeted therapies have been approved for breast cancer, including the monoclonal antibodies trastuzumab and pertuzumab [85,88]. Trastuzumab binds to domain IV of the HER2 extracellular domain and prevents ligand-independent HER2/HER3 signaling, whereas pertuzumab binds to domain II of HER2, preventing ligand-induced HER2/HER2 homodimerization [89,90]. Both trastuzumab and pertuzumab can bind to HER2 without competing with each other, and in fact, they may have a synergistic effect [91]. Trastuzumab became the first-line therapy for HER2+ advanced GC after the phase III ToGA trial [47]. In the ToGA trial, patients with unresectable GC/GEJC were treated with either trastuzumab and chemotherapy or chemotherapy alone. The study found an increased OS of 13.8 months versus 11.1 months (*p* = 0.00046) in patients treated with trastuzumab and chemotherapy [47]. A post hoc analysis of the ToGA trial of patients with higher HER2 status (as defined by IHC 2+/FISH-positive or IHC 3+) found a 4.2-month improvement in OS with trastuzumab [92].

In preclinical models, trastuzumab has been shown to increase HER2 internalization and cross-presentation by dendritic cells, which then stimulates T-cell responses targeting HER-2 [93,94]. Trastuzumab also has other antitumor effects on the immune system, such as the induction of tumor-infiltrating lymphocytes [95]. Thus, studies have investigated the effect of HER2-targeted therapy combined with immunotherapy for the treatment of advanced GC. In KEYNOTE-811, patients with HER2+ advanced GC/GEJC either received pembrolizumab or placebo in combination with trastuzumab and chemotherapy [96]. The addition of pembrolizumab resulted in a significant improvement in the objective response rate (difference of 22.7%, *p* = 0.00006) and the induction of a complete response in some patients (11% versus 4%) [96].

Trastuzumab deruxtecan (T-DXd) is an antibody–drug conjugate (ADC) that has recently been approved for the treatment of metastatic HER2+ breast cancer [97]. It consists of trastuzumab bound to deruxtecan, a cytotoxic topoisomerase I inhibitor, via a cleavable synthetic tetrapeptide-based linker. Once the ADC binds to its target (in this case, HER2), receptor-mediated endocytosis occurs, and the cytotoxic drug is released into the tumor cell. Neighboring-cell death can also occur through drug diffusion and the release of damage-associated molecular patterns into the TME, causing an immune response [98]. The DESTINY-Gastric01 trial compared T-DXd to chemotherapy in patients with HER2+ advanced GC and found improved survival and overall response rates in the T-DXd arm [99]. However, the side effects included interstitial lung disease and myelosuppression [99]. The FDA approved T-DXd as a second-line therapy for HER2+ locally advanced or metastatic GC/GEJC in 2021 [100].

In terms of HER2+ blockade for LAGC, the PETRARCA trial studied the effect of perioperative trastuzumab, pertuzumab, and FLOT compared to FLOT alone in patients with cT2-4 and/or N+ GC/GEJC and HER2+ overexpression [101]. The pCR rate was significantly improved in the trastuzumab/pertuzumab group compared to that in the group undergoing FLOT monotherapy (35% versus 12%, *p* = 0.02), as was the rate of pathologic lymph node negativity (65% versus 39%) [101]. However, the trastuzumab/pertuzumab group experienced more grade 3 or greater adverse events: notably, diarrhea and leukopenia [101]. Ultimately, the negative results of the JACOB trial, which evaluated the efficacy of dual-HER2 therapy and chemotherapy in advanced GC/GEJC, led to a premature termination of the PETRARCA trial [102].

The phase II INNOVATION trial is currently underway for patients with resectable GC/GEJC [103]. This trial is investigating whether neoadjuvant and adjuvant trastuzumab or trastuzumab and pertuzumab in combination with FLOT is more efficacious than chemotherapy alone [103]. Preliminary results are expected in 2023, and the estimated study completion date is in 2028.

## 7. Claudin 18.2: A Newer GC Target

Claudin 18 isoform 2 (CLDN 18.2) is a tight junction protein in the claudin family that is critical in maintaining the integrity of epithelial barriers in the gastric mucosa [104,105]. Prior studies have shown that CLDN 18.2 is aberrantly expressed in a variety of cancers, including GC [106]. It is thought that CLDN 18.2, which is normally embedded in the tight junction, becomes exposed to the gastric lumen in malignant cells [107]. This feature makes CLDN 18.2 an attractive therapeutic target in GC. Interestingly, CLDN 18.2 expression has not been found to be correlated with other prognostic markers, such as HER2 and PD-L1 status [108].

Several clinical studies have been designed to study the safety and efficacy of CLDN-18.2-targeted therapies, such as zolbetuximab (a monoclonal antibody), in advanced GC. The FAST trial, in which patients with CLDN 18.2+ advanced GC/GEJC received zolbetuximab and chemotherapy versus chemotherapy alone, showed significant improvement in PFS and OS (OS in patients who received zolbetuximab and chemotherapy HR = 0.55, *p* = 0.0005) [109]. The SPOTLIGHT phase III trial similarly compared zolbetuximab and leucovorin, fluorouracil, and oxaliplatin (FOLFOX) to FOLFOX alone in patients with advanced GC/GEJC and found preliminary positive results for the zolbetuximab arm. The median PFS was improved with zolbetuximab (10.61 versus 8.67 months, HR = 0.751, *p* = 0.0066) as was the median OS (18.23 versus 15.54 months, HR = 0.75, *p* = 0.0053) [110]. Preliminary results from the GLOW trial have validated the results of the SPOTLIGHT trial [111]. This trial found that zolbetuximab in combination with capecitabine and oxaliplatin (CAPOX) extended OS in patients with unresectable locally advanced or metastatic CLDN 18.2+ GC compared to placebo and CAPOX (14.4 versus 12.2 months, HR = 0.771, *p* = 0.0118) [111].

Autologous T cells that express chimeric antigen receptors (CAR T cells) have arisen in the field of oncology as a means of individualizing cancer treatment and may have utility in CLDN 18.2+ GC. CAR T cells are generated from an individual’s own immune system and are genetically designed to express specific antigen-binding domains that can bind to tumor antigens, such as CLDN 18.2, and amplify these T cells [112]. CAR T cells have been effective in liquid tumors (e.g., leukemia and lymphoma), though they have had limited efficacy in solid tumors [113]. Phase I interim results from a single-arm clinical trial (NCT03874897) in patients with CLDN 18.2+ GC found that CLDN-18.2-specific CAR T cells are safe and have promising efficacy, with a 6-month OS rate of 81.2% and an overall response rate of 57.1% [114]. The final results of this trial are estimated to be released in 2024.

Currently, there are no randomized clinical trials investigating the efficacy of CLDN-18.2-targeted therapies in LAGC, likely because CLDN 18.2 is a relatively new therapeutic target. However, given the positive phase III trials in advanced GC, there may be a role for incorporating CLDN-18.2-targeting therapy in LAGC treatment. This may involve immunological techniques that are currently in development such as monoclonal antibodies, CAR T cells, bi-specific antibodies, antibody–drug conjugates, and others [115].

## 8. FGFR2 Blockade in Advanced GC

The fibroblast growth factor receptor (FGFR) family has four types, FGFR-1, FGFR-2, FGFR-3, and FGFR-4, each with their own substrate-binding selectivity and tissue distribution [116]. When FGFRs are activated, they dimerize and autophosphorylate and then activate downstream signaling pathways such as RAS-RAF-MEK-ERK and PIK3CA-AKT-mTOR, which can lead to cell proliferation and differentiation [117]. Amplification of the FGFR gene has been reported in a variety of cancers; FGFR1 mutations, FGFR2 amplifications and overexpression of the FGFR2b splice variant, and FGFR3 rearrangements are the most common alterations found in GC and can co-occur [118]. FGFR2 overexpression in GC has been associated with a poorer prognosis and response to chemotherapy [119,120].

Bemarituzumab is a novel FGFR2b monoclonal antibody that binds the extracellular domain of FGFR2b, inhibits its signaling, and enhances antibody-dependent cellular cytotoxicity against FGFR2b-expressing tumor cells [121]. The phase II FIGHT trial was the first RCT of an FGFR inhibitor in GC patients [122]. They treated patients with HER2-negative, FGFR2b-selected advanced GC/GEJC with either bemarituzumab or placebo with FOLFOX [122]. They found that bemarituzumab did improve PFS compared to placebo, but this difference was not statistically significant (9.5 versus 7.4 months, *p* = 0.073) [122]. Notably, corneal adverse events (e.g., dry eye, keratitis, corneal erosion) occurred in 67% of patients who received bemarituzumab compared to 10% in the placebo arm [122].

As of now, no trials are investigating FGFR2 blockade in LAGC. However, the phase III trials FORTITUDE-101 and FORTITUDE-102 are planned for the evaluation of the safety and efficacy of bemarituzumab with FOLFOX (FORTITUDE-101) and bemarituzumab with nivolumab and FOLFOX (FORTITUDE-102) in patients with advanced GC/GEJC [123,124]. These studies are expected to be completed in 2025 and 2027, respectively. FGFR2 blockade is another promising therapeutic target for the treatment of advanced GC and LAGC, though more clinical studies are needed.

## 9. Future Directions

GC is being divided into an increasing number of subtypes based on molecular profiling and targetable pathways. New therapeutics introduced in the treatment of advanced GC, such as immunotherapy and HER2 blockade, have paved the way for tailored treatment strategies for LAGC. Given that more than half of patients with LAGC may experience disease recurrence after surgical resection, developing new and improved neoadjuvant and adjuvant therapies remains critical [16]. Table 1 summarizes the major recent and ongoing phase II/III trials of LAGC adjuvant therapies that have been discussed in this review article.

Figure 1 illustrates an example of a possible, future, biomarker-driven trial for LAGC, with cohorts delineated by clinical biomarkers of interest and treatments differing based on the cohort. Our increasing understanding of molecular subtypes and biomarkers will likely continue to revolutionize the treatment of LAGC. In the future, LAGC patients with resectable LAGC will likely be stratified into an increasing number of groups based on key biomarkers discussed in this review article and will undergo combined chemotherapy, immunotherapy, and biomarker-directed therapy before and after surgical resection.

## 10. Conclusions

This review highlights the novel biomarkers (MSI status, PD-L1 CPS, HER2, CLDN 18.2, FGFR2) utilized in the treatment of LAGC and discusses recent and ongoing clinical trials shaping the landscape of LAGC perioperative therapy. As clinical trials continue to clarify the roles of chemotherapy, immunotherapy, and targeted molecular therapies in the perioperative period, the timing and extent of curative resection will likely evolve. Precision medicine is likely the future of oncology, and as our knowledge of the tumor biology of LAGC grows, so will our ability to prevent recurrence after resection and to combat treatment resistance.

## Figures and Tables

**Figure 1 cancers-15-04114-f001:**
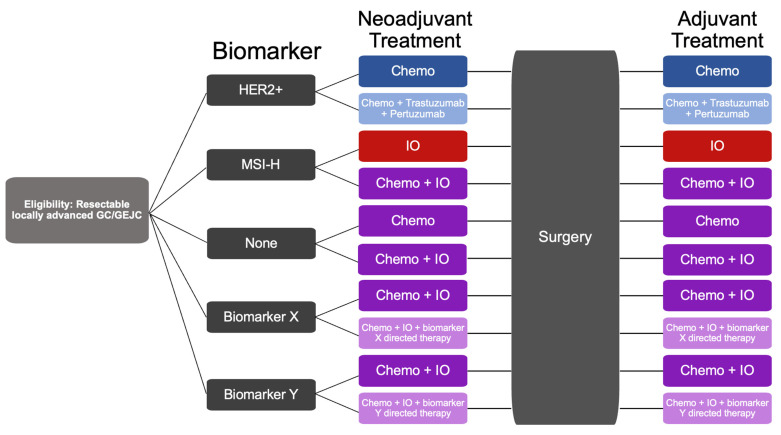
Schematic of a future neoadjuvant biomarker-driven clinical trial for LAGC. Patients with resectable, locally advanced, GC/GEJC will be stratified into cohorts based on biomarkers: HER2+, MSI-H, biomarker X, biomarker Y, or none of the above. Biomarker X and Y are placeholders for current (such as FGFR or CLDN 18.2) or future biomarkers of clinical interest in LAGC. Each biomarker cohort will have two differing treatment arms based on data from prior clinical trials. Patients will undergo combination neoadjuvant chemotherapy, immunotherapy, and/or biomarker-directed treatment for 4–6 months and then undergo surgical resection followed by adjuvant therapy based on their cohort. The primary outcome is the achievement of 30% pCR. IO = immuno-oncology therapy. Chemo = chemotherapy.

**Table 1 cancers-15-04114-t001:** Select completed or ongoing phase II/III trials for LAGC.

Clinical Trial Information	Arms	Patient Population	Status	Primary Outcomes
NCT04006262 GERCOR NEONIPIGA Phase II	Arm 1: Perioperative nivolumab plus neoadjuvant ipilimumab	cT2-T4/NX/M0 resectable GC/GEJC, dMMR/MSI-H+	Recruiting	pCR rate 58.6%
NCT04817826 INFINITY Phase II	Arm 1: Neoadjuvant tremelimumab and durvalumab	MSI-H+, c ≥ T2, any N, any M, GC/GEJC	Recruiting	Overall pCR rate 60%, T2-3 tumor pCR rate 89%
NCT03421288 DANTE Phase II	Arm 1: Perioperative atezolizumab and FLOT ^1^ Arm 2: Perioperative FLOT	c ≥ T2 or N+ GC/GEJC	Active, not recruiting	PFS—pending data
NCT03221426 KEYNOTE-585 Phase III	Arm 1: Perioperative pembrolizumab and XP ^2^/FP ^3^/FLOT Arm 2: Perioperative XP/FP/FLOT	Stage II-IVa GC/GEJC	Active, not recruiting	Event-free survival, pCR, OS, rate of adverse events—pending data
NCT04592913 MATTERHORN Phase III	Arm 1: Perioperative durvalumab plus FLOT Arm 2: Perioperative FLOT	≥Stage II GC/GEJC	Active, not recruiting	Event-free survival—pending data
NCT04250948 Phase II	Arm 1: Perioperative toripalimab and SOX/XELOX ^4^ Arm 2: Perioperative SOX/XELOX	Resectable cT3-4a/N+/M0 GC/GEJC	Recruiting	Pathological complete regression/moderate regression rate (TRG 0–1): Arm 1 44.4% vs. Arm 2 20.4%, *p* = 0.009
NCT02581462 PETRARCA Phase II	Arm 1: Perioperative trastuzumab and pertuzumab and FLOT Arm 2: Perioperative FLOT	cT2-T4 and/or N+ GC/GEJC with HER2 overexpression	Completed	pCR: Arm 1 35% vs. Arm 2 12%, *p* = 0.02
NCT02205047 INNOVATION Phase II	Arm 1: Perioperative trastuzumab and FLOT Arm 2: Perioperative trastuzumab, pertuzumab, and FLOT Arm 3: Perioperative FLOT	Resectable GC with HER2 overexpression	Active, not recruiting	Major pathological response rate (<10% vital residual tumor cells)—pending data

^1^ FLOT: 5-fluorouracil, leucovorin, oxaliplatin, docetaxel; ^2^ XP: capecitabine and cisplatin; ^3^ FP: 5-fluorourical and cisplatin; ^4^ SOX/XELOX: S-1 and oxaliplatin/capecitabine and oxaliplatin.

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
