# Peer review of "The Present and Future of Neoadjuvant and Adjuvant Therapy for Locally Advanced Gastric Cancer"

_cancers, 2023, doi:10.3390/cancers15164114_

Round 1
Reviewer 1 Report
The authors described the present treatments and introduced evidences about advanced gastric cancer treatment.
This article is a narrative review of neoadjuvant and adjuvant chemotherapy for locally advanced gastric cancer (LAGC).
The authors described about the evidences of these topics, especially from western countries.
They also described about on-going studies, mentioning to future perspectives of this fields.
Since this is not a systematic review, I have no comments about the methodology.
I thought references are appropriate, and conclusion that precision medicine using some biomarkers is likely the future of oncology seems to be interesting and beneficial for readers.
(Minor comments)
"Biomarker", "Neoadjuvant therapy", and "Adjuvant therapy" should be used as an index of the Figure 1.
Reviewer 2 Report
The authors present a interesting and well written review of neoadjuvant and neoadjuvant therapy for locally advanced gastric cancer. The manuscript is pertinent due to the poor results obtained with the current cytotoxic chemotherapy in this cancer and the amount of data published about the new biologic agents.
The paper is well clear, presented and discused. I have no queries.
1. The objective is clearly presented and discused.
2. The text of the review is clearly developed, data are well clasified, easy to read and very informative. For me, the points about "Molecular subtypes and biomarkers of GC" and "Immune checkpoint blockade" have provided interesting information.
3. Table and Figure and apropiate.
4. References are apropiate (124 references)
1. The present manuscript have reviewed the up-to-date results of adjuvant and neoadjuvant in locally advanced gastric cancer.
2. The topic is specially relevant, it address a gap in the actual treatment of advanced gastric cancer. The results obtained with the actual chemoterapeutic agents is very poor. But their is a growing number of publications about the introduction of novel biologic agents. The results are promising but dispersed. The results obtained with the new biologic agents needs to be systematized and reviewed.
3. This review provide up-to-date information, and provide a clinical interpretation of the data. 4. The authors considers that there are promising results, but ongoing Clinical trials need to clarify the roles of chemotherapy, immunotherapy, and targeted molecular therapies in the treatment of these cancers. 5. The conclusions are consistent with the data presented in the manuscript. 6. Table and Figure are apropiate.
